# 3D printing colloidal crystal microstructures via sacrificial-scaffold-mediated two-photon lithography

Keliang Liu[1], Haibo Ding[1], Sen Li[1], Yanfang Niu[1], Yi Zeng ◉[1], Junning Zhang[1], Xin Du ◉[1] ✉ & Zhongze Gu ◉[1] ✉

The orderly arrangement of nanomaterials' tiny units at the nanometer-scale accounts for a substantial part of their remarkable properties. Maintaining this orderness and meanwhile endowing the nanomaterials with highly precise and free-designed 3D micro architectures will open an exciting prospect for various novel applications. In this paper, we developed a sacrificial-scaffold-mediated two-photon lithography (TPL) strategy that enables the fabrication of complex 3D colloidal crystal microstructures with orderly-arranged nanoparticles inside. We show that, with the help of a degradable hydrogel scaffold, the disturbance effect of the femtosecond laser to the nanoparticle self-assembling could be overcome. Therefore, hydrogel-state and solid-state colloidal crystal microstructures with diverse compositions, free-designed geometries and variable structural colors could be easily fabricated. This enables the possibility to create novel colloidal crystal microsensing systems that have not been achieved before.

Colloidal crystals can produce unfading macroscopic structural colors by arranging colloidal nanoparticles in a periodic manner[1–3], which generates a forbidden gap that prevents the light of a specific wavelength from propagating through the material[4,5]. Because of this unique property, colloidal crystals are widely employed in the preparation of novel optoelectronic devices[6], information carriers[7], and sensing devices[8–11]. Typically, colloidal crystal materials are fabricated as millimeter/centimeter scale films and highly precise microstructures—especially in the Z direction—is almost inhibited. This significantly limits the application of these smart materials as complex microdevices and microsensing systems. Although fabrication strategies such as lithography[12], inkjet printing[13], and microfluidic synthesis[14] can endow the colloidal crystal materials with sub-milimiter geometries to some extent, constructing complex 3D colloidal crystal microstructures with orderly-arranged nanoparticles is still a great challenge for the state-of-the-art technologies.

Generally, two types of strategies are normally employed to fabricate 3D photonic crystal materials. One is the bottom-up fabrication strategy, namely, building large-scale ordered structures via the natural self-assembly of nanoparticles[15–18]. By using nanoparticle-containing inks, macroscopic 2D or simple 3D structures composed of nanoparticle aggregates could be generated[19,20]. However, the resolution, material type, and freedom of styling in the Z direction of the obtained structures are quite limited. The other strategy is the top-down material molding strategy[21]. Utilizing an artificial technology, two-photon lithography (TPL), free-designed photonic crystal materials with highly-ordered 3D nanostructures can be fabricated from a photoresist[22,23]. Although this method exhibits many advantages compared to the bottom-up strategy, creating forbidden gaps at visible-light range requires a high resolution that is beyond the normal TPL system, and the type of available photoresin is quite limited.

We believe that the disadvantages of both bottom-up and top-down strategies can be overcome by combining them: use TPL as the fabrication method and nanoparticle-containing ink as the photoresist. By this way, TPL generates macroscopic objects with micrometer-scale substructures and the homogenously dispersed nanoparticles endows the objects with the structural color feature. In fact, the combination of the TPL and colloidal particle self-assembly has been demonstrated in

[1]State Key Laboratory of Bioelectronics, School of Biological Science and Medical Engineering, Southeast University, Nanjing 210096, China.
✉e-mail: du.xin@seu.edu.cn; gu@seu.edu.cn

some previous works, which enable the possibility to guild the alignment of nematic liquid crystals with colloidal particle geometries[24–26], or generate inverse opal photonic crystal microstructures with colloidal particle templates[27]. Unfortunately, as we tested, hydrogels generated following this way exhibit no structural colors, since the laser writing process would disturb the stable microenvironment in the photoresist, which is necessary for nanoparticle self-assembling (see in discussion part).

To overcome this challenge, we developed a strategy, which was named as "sacrificial-scaffold-mediated TPL", to conduct TPL fabrication in nanoparticle-containing photoresists. By constructing a degradable hydrogel scaffold (sacrificial scaffold) to "lock" the self-assembled nanoparticles, the disturbance effect of the femtosecond laser during TPL fabrication can be minimized. Therefore, solid-state or hydrogel-state micro-objects with precise and highly-free 3D geometries as well as iridescent structural colors were fabricated successfully. We show that the minimum feature size of the generated colloidal crystal microstructures can go down to 3 µm, and the structural color of the printed microstructures can be well controlled by the TPL processing parameters. Our strategy is available for diverse monomers, including those incompatible with nanoparticles. This makes the method a powerful tool for the fabrication of functional colloidal crystal devices for different applications. We demonstrate such potential by generating microsensors for in-situ temperature monitoring in microfluidic devices.

## Results

### Colloidal crystal sacrificial scaffold

Typically, a colloidal crystal film (hydrogel as an example) can be easily generated by polymerizing a precursor solution containing monomer, solvent, and monodispersed nanoparticles under an undisturbed atmosphere. In this case, the nanoparticles are arranged periodically in the formed hydrogel, leading to a photonic band gap that reflects light at a specific wavelength[11]:

$$\lambda = 2\sqrt{\frac{2}{3}}d\sqrt{n_a^2 f + n_b^2(1-f) - \sin^2\theta} \qquad (1)$$

where $d$ is the center distance of the adjacent silica particles, $n_a$ and $n_b$ are the refractive indices of silica particles and hydrogel, respectively, $f$ is the volume fraction of the structured material and $\theta$ is the angle of incidence. This was confirmed by our test, as shown in Supplementary Fig. 1a, a colloidal crystal hydrogel film with blue structural color can be easily obtained by curing an aqueous precursor containing polyethylene glycol diacrylate (PEGDA, the refractive index of the cured hydrogel matrix is 1.427) and silica nanoparticles (Φ = 150 nm, refractive index 1.46) under the UV irradiation. However, when processing the precursor solution directly through TPL, this method became ineffective, the obtained micro hydrogel did not show any structural color (an example is shown in Supplementary Fig. 1b). SEM analysis on the micro hydrogel indicated that nanoparticles were disorderly arranged (Supplementary Fig. 1c, d), which was contrary to the hydrogel film cured by UV. This phenomenon could probably be attributed to the disturbance effect of the femtosecond laser. During the TPL fabrication, the laser focus is drawn across the precursor solution to polymerize the monomers. In this process, the optical gradient force generated by the high-energy density femtosecond laser (see Supplementary Fig. 2 for more information) caused the nanoparticles to be temporarily bound in the focal point, which may disturb their ordered arrangement[28,29]. In addition, the shear force generated by the relative movement of the solidified precursor and the surrounding unpolymerized liquid would cause the nanoparticles to move towards the trajectory of laser scanning, which also significantly disturbs the assembly of the nanoparticles in the precursor solution[30].

To address this problem, we developed a sacrificial-scaffold-mediated TPL method to reduce the disturbance effect of the femtosecond laser (Fig. 1). The key of the method is to generate a degradable hydrogel scaffold (sacrificial scaffold) that can "lock" the position of the nanoparticles during TPL process. To do this, a solution containing bis(2-methacryloyl)oxyethyl disulfide (BMOD) and monodispersed nanoparticles (Φ = 180 nm) in dimethylformamide was formed after 4 min illumination under 365 nm UV (70 mW/cm²), a colloidal crystal hydrogel with bright green structural color was obtained (Fig. 2bi). SEM analysis on the freeze-dried hydrogel film confirmed the ordered distribution of nanoparticles inside the hydrogel. Such sacrificial scaffold contained a polymer network with high density disulfide crosslinkages (Supplementary Fig. 3a), capable of "locking" the inner nanoparticles during TPL fabrication, to minimize the disturbance effect from femtosecond laser. After TPL processing, the scaffold could be degraded by dissociating the disulfide bond with tris(2-carboxyethyl)phosphine (TCEP)[31,32], to remove the unprinted area, leaving TPL-fabricated 3D microstructure with orderly-arranged nanoparticles inside (Fig. 1).

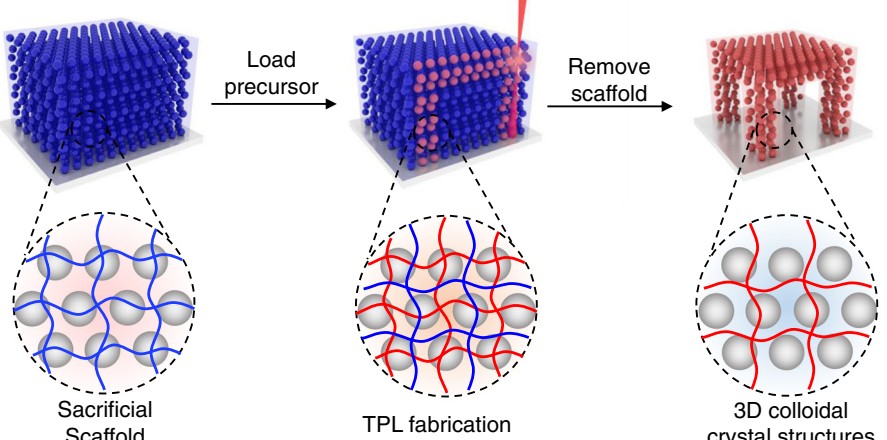

Load precursor

Remove scaffold

Sacrificial Scaffold

TPL fabrication

3D colloidal crystal structures

**Fig. 1 | Schematic demonstration of the sacrificial-scaffold-mediated TPL process.** The sacrificial scaffold containing a degradable hydrogel network and orderly-arranged nanoparticles is first generated. Then hydrogel precursor is introduced into the scaffold via solvent exchange process, and TPL fabrication was performed. Finally, the hydrogel is immersed in a specific solution to degrade the sacrificial scaffold, leaving the TPL-fabricated hydrogel microstructures with inner nanoparticles.

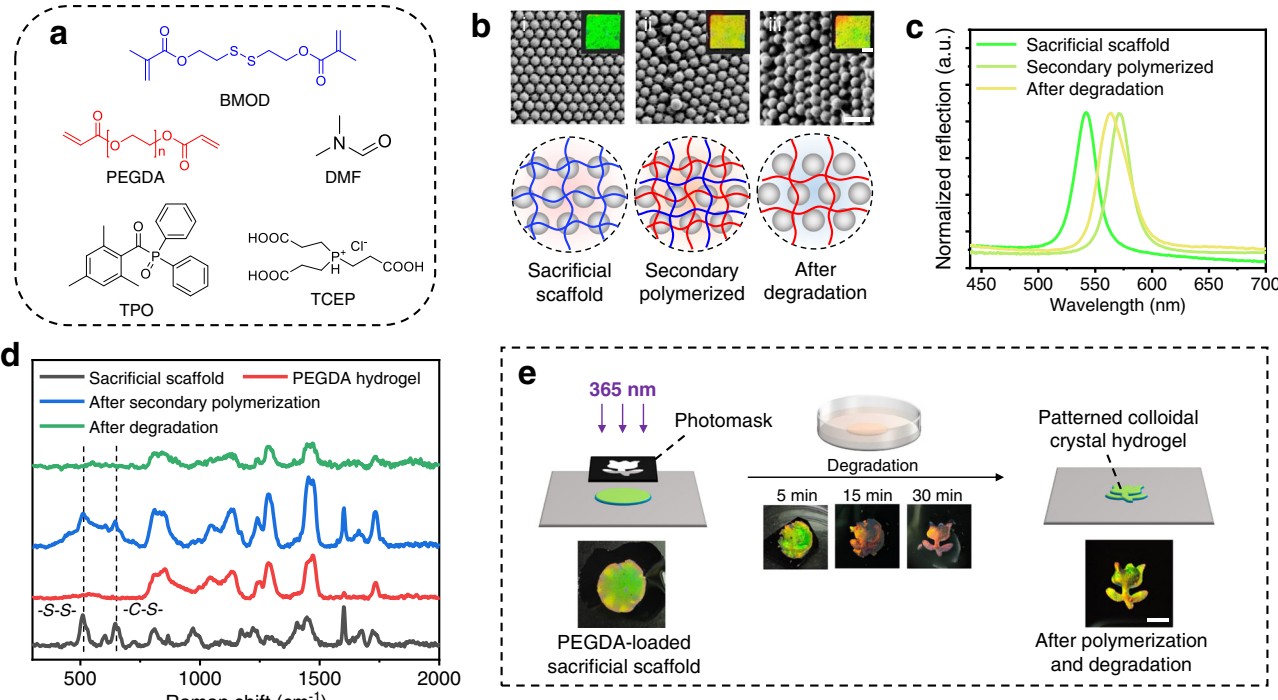

**Fig. 2 | Colloidal hydrogel fabrication using the sacrificial scaffold. a** Chemical structure of the reagents used. **b** Optical and SEM images of the (i) sacrificial scaffold, (ii) secondary polymerized hydrogel, and (iii) hydrogel after degradation. Scale bars: 2 mm for optical images and 500 nm for SEM images. **c** Normalized reflection spectra of the hydrogels in **b**. **d** Raman spectrum of the PEGDA hydrogel (fabricated without sacrificial scaffold), and the hydrogels in **b**. **e** Schematic demonstration of the formation of patterned PEGDA colloidal crystal hydrogel using a sacrificial scaffold, and the photo of the hydrogels obtained during the fabrication process. Scale bars: 5 mm.

The effect of the sacrificial scaffold was first investigated at the macroscopic scale. PEGDA precursor solution was introduced into the sacrificial scaffold through the solvent exchange, and the hydrogel film was then exposed to 70 mW/cm$^2$ UV to form a double-network hydrogel film (Fig. 2bii and Supplementary Fig. 3b). Afterwards, the hydrogel was immersed in TCEP solution to remove the sacrificial scaffold by reducing disulfide bonds to mercapto groups (Supplementary Fig. 3c). A structural-colored hydrogel (Fig. 2biii) was obtained rather than completely degraded. The reflection spectra of the hydrogel film at each state were analyzed with a fiber optic spectrometer, as shown in Fig. 2c. The spectra exhibit clear reflection peaks, indicating that the nanoparticles inside the hydrogels are periodically arranged. This was also confirmed by SEM, that the ordered arrangement of the silica particles in the sacrificial scaffold only slightly changed after the secondary polymerization and degradation process (Fig. 2b). It's notable that the color and reflection peak of the hydrogel film slightly red-shifted after PEGDA polymerization and the degradation process (Fig. 2c), this might be attributed to the synergistic effect of the inter-particle packing of the secondary polymerized gel network and the further swelling of the film in the liquid environment, resulting in an increase in the nanoparticle spacing (*d* in Eq. 1). To track the chemical composition of the hydrogel film at each state, Raman spectrometry was performed. As shown in Fig. 2d, the sacrificial scaffold exhibits characteristic peaks of S-S bond (~515 cm$^{-1}$) and C-S bond (~645 cm$^{-1}$), after the secondary polymerization process, the hydrogel shows a spectrum which is the overlay of sacrificial scaffold and pure PEGDA hydrogel, confirming its double-network composition. After the degradation process, the characteristic peaks of the S-S and C-S bond disappeared, and the obtained spectrum is highly close to that of pure PEGDA hydrogel. This proved our hypothesis that the polymer network of the sacrificial scaffold is completely removed from the hydrogel after degradation, leaving the secondary polymerized hydrogel network with orderly-arranged nanoparticles inside. To further confirm the possibility of fabricating hydrogel patterns in the sacrificial scaffold, a photomask with a "tulip" pattern was applied during the secondary polymerization process (Fig. 2e). The results showed that, after 30 min degradation, a tulip-shaped colloidal crystal hydrogel film was obtained, while the surrounding sacrificial scaffold disappeared completely. This demonstrated the ability to generate colloidal crystal hydrogels with designed geometries using the sacrificial scaffold.

## Sacrificial-scaffold-mediated TPL

Then, we applied our strategy to the TPL process to fabricate 3D colloidal crystal microstructures. After loading PEGDA precursor solution, the sacrificial scaffold was placed into a commercial TPL system with 780 nm femtosecond laser (Nanoscribe GT$^+$). The moving path of the laser focus is obtained by model slicing software (Nanowrite), and based on this path, a microcube model is constructed in the colloid crystal film by point scanning (Fig. 3ai). After TPL fabrication, a green cube was observed in the sacrificial scaffold (Fig. 3aii). By a further degradation process, the sacrificial scaffold was completely removed, and a hydrogel microcube was obtained (Fig. 3aiii, iv). The generated hydrogel microcube exhibits a clear red-shifted reflection peak (Fig. 3b), which is close to the result in the macroscopic experiment (Fig. 2c), indicating the well-ordered nanoparticle arrangement inside the hydrogel. To confirm this, SEM analysis was performed on the top, cross-section and bottom of the freeze-dried hydrogel microcube (Fig. 3ci). A well-ordered arrangement of the nanoparticles could be observed on the cross-section and bottom of the micro hydrogel, this proves that the sacrificial scaffold can efficiently reduce the disturbance effect of the femtosecond laser. On the top side of the hydrogel, the nanoparticles seem not well-ordered, this might be because the top surface of the hydrogel is not as flat as the cross-section (cut) and bottom surface (aligned by the flat glass substrate).

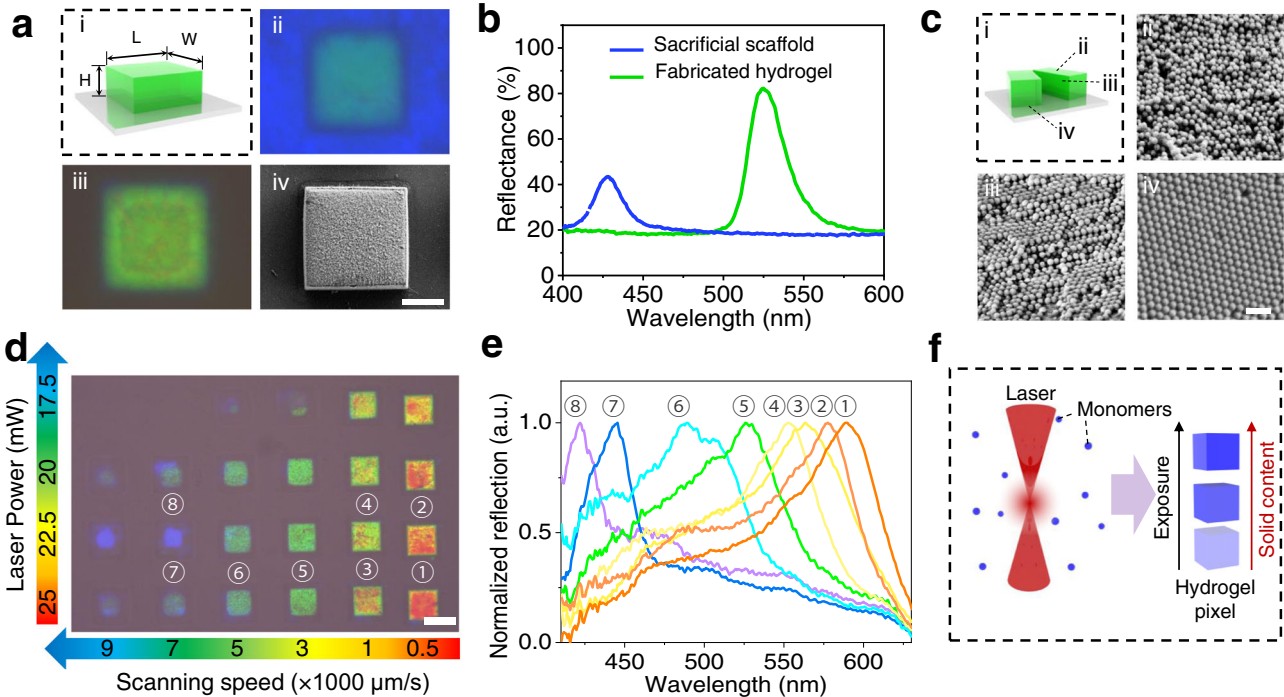

**Fig. 3 | Characterization of the sacrificial-scaffold-mediated TPL process.**
**a** Colloidal hydrogel microcube fabricated with the method. (i) Model of the microcube (L = W = 50 μm, H = 25 μm). (ii) Photo of the microhydrogel before degradation process. The blue area is the sacrificial scaffold and the green area is the inner fabricated PEGDA hydrogel cube. (iii) Photo of the microhydrogel after degradation process. (iv) SEM image of the fabricated hydrogel cube. Scale bars: 20 μm. **b** Reflection spectra of the sacrificial scaffold and fabricated microhydrogel. **c** SEM images of the hydrogel microcube from (ii) top, (iii) cross-section and (iv) bottom (Φ = 180 nm, Scale bars: 500 nm). **d** Microcube array fabricated with gradient-varied laser power and scanning speed, demonstrating the dependence of the obtained structure color with the TPL processing parameters. Scale bars: 50 μm. **e** Reflection spectra of the selected microcubes in **d**. **f** Proposed mechanism of the exposure-dependence of the structure color of the fabricated hydrogel microcubes.

After confirming the feasibility of the sacrificial-scaffold-mediated TPL, we investigate the effect of the processing parameters (laser power and scanning speed) on the structural color of the obtained hydrogels. A 6 × 4 microcube array (50 × 50 × 30 μm for each cube) was fabricated in the PEGDA-loaded sacrificial scaffold, the laser scanning speed (500–9000 μm/s) and laser power (17.5–25 mW) during fabrication were gradually varied. After removal of the sacrificial scaffold, the obtained array was observed under the microscope. Interestingly, as shown in Fig. 3d and Supplementary Fig. 4, we found that the structural color of the microcube red shift with a higher exposure dose (increase of laser energy, or decrease of scanning speed). The reflection peak of the microcubes redshift from 420 nm to 590 nm with higher exposure doses (Fig. 3e), which also confirm the results. This implies that it is possible to fabricate micro hydrogels with diverse structural colors from one sacrificial scaffold, making our strategy quite versatile. We believe that a larger exposure dose would increase the polymerization degree of the precursor solution, leading to hydrogel voxels with higher solid content (Fig. 3f), thus enlarging the distance between the nanoparticles and finally inducing a redshift of the reflection peak. To verify this hypothesis, we performed a similar experiment at the macroscopic scale, the PEGDA-loaded sacrificial scaffold was irradiated under 365 nm UV LED for 3 min. To significantly increase the solid content of the hydrogel, the precursor solution in the sacrificial scaffold was renewed before prolonging the irradiation at each time point (Supplementary Fig. 5). As we expected, the color of the hydrogel film red shift from green to red and then infrared after each exposure. Therefore, the color change of the fabricated hydrogels could probably be attributed to the change of solid content of the hydrogel voxels under different TPL processing parameters.

## Fabrication of 3D colloidal crystal microstructures

Owing to the high resolution and high degree of freedom of the TPL fabrication method, structure-colored micro hydrogels with diverse 2D and 3D geometries could easily be generated using our method. To demonstrate this, we present various hydrogel microstructures fabricated using our strategy. As shown in Fig. 4a, 2D micro hydrogels with diverse geometries and structural colors were successfully fabricated. The resolution of the method was tested by generating hydrogel lines with different line width (1 μm to 20 μm), the result (Supplementary Fig. 6) shows that the minim line width that can be achieved by our method is at least 3 μm. The reproducibility of the method was analyzed by repeatedly fabricating a hydrogel microcube (50 × 50 × 30 μm) for 54 times in a sacrificial scaffold and measuring the hue value of each microcube after degradation. As shown in Fig. 4b, the obtained microcubes shows identical structural colors, with an average hue value of 0.35 ± 0.1, indicating the high reproducibility of this method (another 225 repeat experiment are shown in Supplementary Fig. 7). To demonstrate the ability of our method for the fabrication of complex and precise colloidal crystal hydrogel microstructures, a 3D hollow spiral micropipe was fabricated (Fig. 4ci). As shown by SEM image (Fig. 4cii), a micro hydrogel with ultra-complex 3D structure was successfully obtained. The spiral structure was well maintained after the degradation process, and the hollow pipe structure, as well as the designed micropore on the pipe wall, was clear observed (Fig. 4cii and Supplementary Fig. 8). By varying the processing parameters, micropipes with clear blue, green and red structural colors were generated (Fig. 4ciii). This confirms the ability of our method to fabricate 3D colloidal crystal microstructures with intricate designs.

Controlling the fabrication parameters during TPL processing, multi-color 3D micro hydrogels can be easily generated. As an

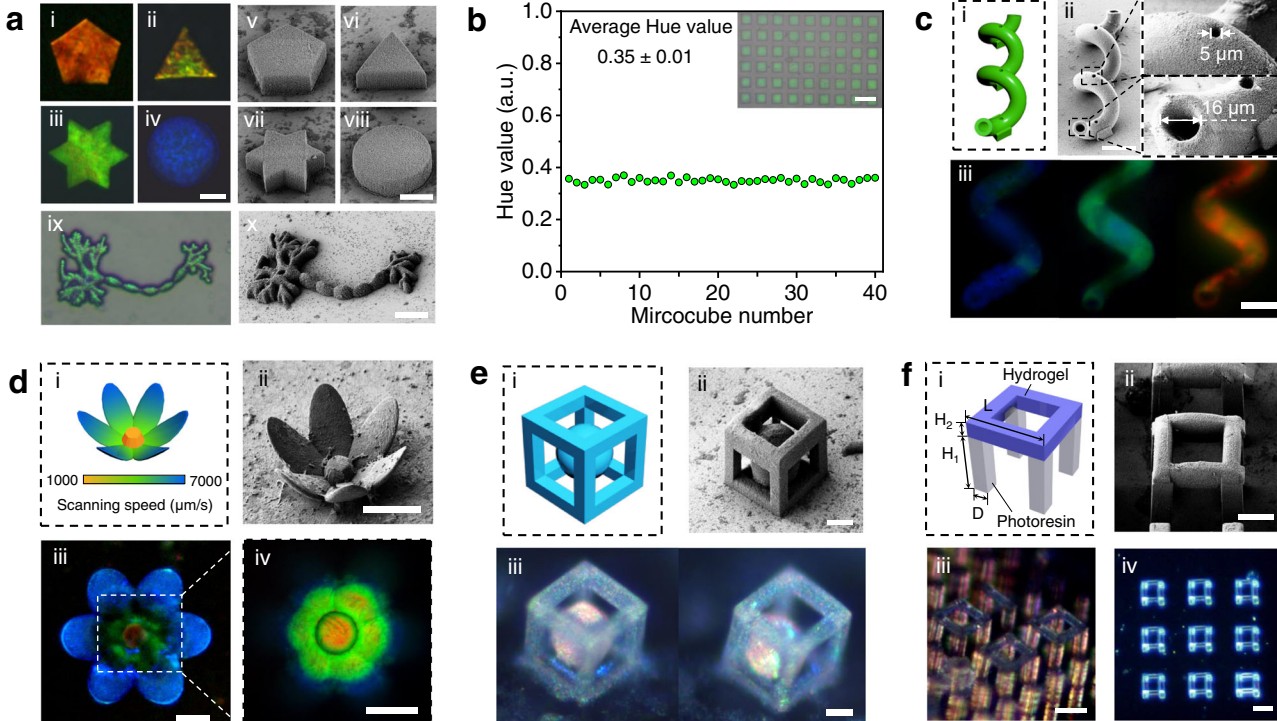

**Fig. 4 | 2D and 3D microhydrogels fabricated via the sacrificial-scaffold-mediated TPL. a** 2D colloidal crystal microstructures with different geometries and colors. (i–iv) Optical images and (v–viii) SEM images. (ix) Optical image of a microhydrogel with neuron cell geometry. (x) SEM image of the freeze-dried hydrogel. Scare bars: 20 μm. **b** Reproducibility test, by printing a hydrogel microcube 54 times and measure their hue value (40 shown in the graph). Scare bar: 100 μm. **c** (i) design, (ii) SEM image and (iii) optical images of 3D hollow spiral microtubes fabricated by our strategy. Scare bars: 40 μm. **d** (i) design, (ii) SEM image and (iii, iv) optical images of a multi-color three-dimensional microhydrogel with lotus-like geometry, constructed by varying laser scanning speed (laser power = 20 mW) during TPL fabrication. Scale bars: 40 μm. **e** A colloidal crystal hydrogel cube frame with an inner suspended sphere. (i) design of the structure, (ii) SEM image of the freeze-dried microhydrogel and (iii) photos of the hydrogel under different conditions, revealing the suspending state of the inner hydrogel sphere. Scale bars: 20 μm. **f** Colloidal crystal hydrogel "bridge" constructed on a micropillar array. (i) Design of the structure, H1 = 40 μm, H2 = 10 μm, D = 10 μm, L = 60 μm. (ii) SEM image of the freeze-dried hydrogel. (iii, iv) Optical images of the microstructure. Scale bars: 20 μm for SEM images and 50 μm for optical images.

example, in Fig. 4d we show a lotus-like 3D microstructure with gradient structural color from stamen to petals. The difference in structural color is caused by the variation of processing parameter (laser scanning speed) during the fabrication of stamen and petal parts. The petals of the obtained hydrogel exhibit bright blue structural color, and the stamen shows green-red color, demonstrating the ability to fabricate muti-color colloidal crystal microstructures by our method.

Another important advantage of the sacrificial scaffold is that the printed hydrogel voxel during the TPL fabrication can be mechanically "held" by the scaffold, therefore, suspended structures, which is difficult or even impossible to be fabricated by normal TPL or other 3D printing technologies, can be directly written by our method without collapsing. As an example, we fabricated a cube frame with a suspended inner sphere (Fig. 4e and Supplementary Fig. 9). In such a structure, the sphere has no connection with the surrounded frame, which is challenging for traditional fabricating strategies. While in our method, the whole structure could be directly written in the sacrificial scaffold. After the degradation process, the obtained microsphere can be freely moved in the cube frame, and both the frame and sphere exhibit very clear structural colors (Fig. 4eiii). In another example, suspended colloidal crystal hydrogel bridges (Fig. 4fi) were written on a solid-state micropillar array (pre-fabricated using a commercial photoresist IP-S). The complex construction and the clear structural color of the hydrogel are confirmed by the SEM analysis and microscopic images. The generation of hierarchical colloidal particles and their assembled arrays were also achieved using our method (Supplementary Fig. 10). In Supplementary Fig. 11, we show photos of

various 2D and 3D colloidal crystal hydrogels fabricated using our strategy, which further proves the capability of our method. Using our approach, photonic crystal hydrogels with arbitrary geometries can be generated on the target position of a substrate, this may enable the potential to construct complex and accurate photonic sensors to monitor the change in motion and parameters of a microdomain in the future.

## Material versatility of the fabricated microstructures

In our strategy, the sacrificial scaffold is formed before loading the hydrogel precursor, therefore, the type of hydrogel precursors would not affect the property of the sacrificial scaffold and the inner orderly-arranged nanoparticles. As a consequence, our method is a general strategy for the microfabrication of various materials. Such potential was tested in macroscopic scale, hydrogel precursors containing acrylic amide (Fig. 5ai), hydroxyl methacrylate (HEMA, Fig. 5aii) and methacrylate gelatin (GelMA, Fig. 5aiii) were used to fabricate colloidal crystal hydrogels directly or with the help of sacrificial scaffold. The results indicate that, with the sacrificial scaffold, hydrogel films with bright structural colors were generated for all precursors. While without the scaffold, hydrogels generated using HEMA and GelMA precursors show very poor structural color, probably because these monomers would somehow interrupt the self-assemble of the nanoparticles. This proves that the sacrificial scaffold can be used to generate colloidal crystal hydrogels with broad types of monomers, in spite of their compatibility with the nanoparticles. Solid-state colloidal crystal materials can be formed in the same manner, as shown

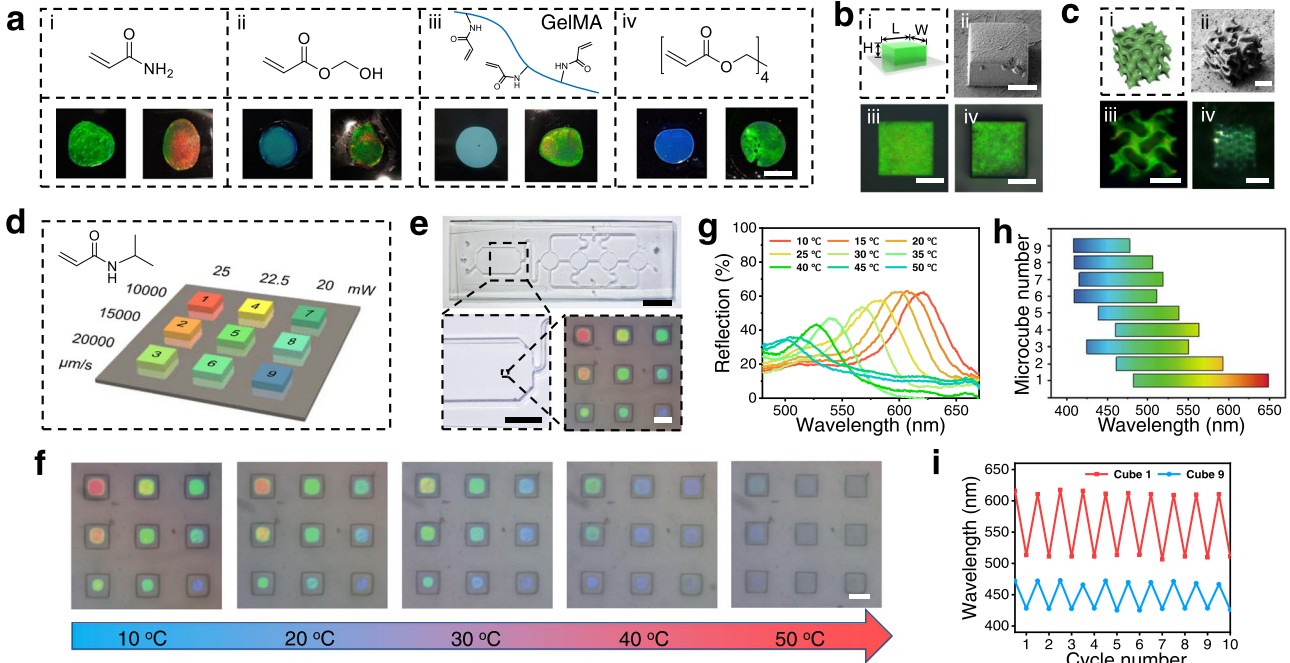

**Fig. 5 | Colloidal crystal materials with various compositions generated with the sacrificial scaffold. a** colloidal crystal hydrogel films fabricated with different precursors. The upper is the structure of the monomers, and the bottom is the fabricated hydrogel film (left: without sacrificial scaffold; right: with sacrificial scaffold). With the sacrificial scaffold, hydrogel films with bright structure color were obtained in spite of the compatibility of the monomers with nanoparticles. Scale bar: 3 mm. **b** Solid-state colloidal crystal microcube fabricated using pure PTTA as the precursor. (i) Model of the microcube (L = W = 50 μm, H = 25 μm). (ii) SEM image of the freeze-dried hydrogel microcube. (iii) Photo of the cube in DMF and (iv) in water. Scale bars: 25 μm. **c** Soild-state colloidal crystal 3D microstructure fabricated with pure PTTA as precursor. (i) Model of the gyroid microstructure. (ii) SEM image of the obtained gyroid microstructure. (iii) Top view of the gyroid microstructure. (iv) 3D view of the object. Scale bar: 20 μm. **d** Processing parameter of the temperature sensor based on polyNIPAAm microhydrogel array. **e** the optical picture of temperature sensors in the microfluidic chip. Scale bar: 1 cm, 0.5 cm and 50 μm. **f** Photo of the hydrogel array under different temperature. Scale bar: 50 μm. **g** Reflective spectra of the hydrogel cube 1 under different temperatures. **h** The maximum range of the reflection peak change of the hydrogel cubes under different temperatures. **i** Change the reflection peak of hydrogel cube 1 and cube 9 when the temperature switches between 20 °C and 40 °C for 10 cycles.

in Fig. 5aiv, when pure pentaerythritol tetraacrylate (PTTA) is used as precursor, a solid film with while-blue color (normally come from the Mie scattering of disorderly-arranged nanoparticles[33]) was obtained. This is probably because silica nanoparticles cannot be homogeneously dispersed in the hydrophobic precursor. While with the help of sacrificial scaffold, colloidal crystal polymer film was formed successfully. This allows us to fabricate solid-state colloidal crystal microstructures with the monomer demanded. To prove this, a microcube (50 × 50 × 30 μm) was fabricated using pure PTTA as the precursor with our method (Fig. 5b). The precise geometry and bright structural color of the obtained microcube are confirmed by the SEM and microscopic images. In contrast to the micro hydrogels above, the structural color of the microcube is quite stable in different solvents or in air (Fig. 5b and Supplementary Fig. 12), demonstrating that the obtained cube is in the solid state rather than in a hydrogel state. Such difference is also confirmed by SEM images (Supplementary Fig. 13). We also tested the effect of the processing parameter to the structural color in this case (Supplementary Fig. 14). However, unlike the situation in hydrogel fabrication, the processing parameter does not affect the color of the obtained solid-state microcubes, which is evidence for the mechanism we proposed in Fig. 3f (as the solid content is always 100% in a solid-state material). 3D micro-object with more complex structures (Fig. 5c) and inverse opal microstructures (Supplementary Fig. 15) were also fabricated, showing the ability of our method to create solid-state colloidal crystal materials with desired geometries.

The broad selection of precursors for the TPL fabrication in our method allow us to fabricate colloidal crystal materials with diverse compositions and geometries, to fulfill the demands from different fields. As a demonstration of the application, we show the

construction of a microsensor system for in-situ temperature monitoring. A precursor solution containing N-isopropylacrylamide (NIPAAm) was loaded to the sacrificial scaffold and then processed with TPL, the scanning speed and laser power during fabrication were specifically controlled (Fig. 5d) to form a 3 × 3 microcube array. Owing to the difference in processing parameter, these microcubes exhibits distinguish structural colors. Since poly(NIPAAm) exhibits temperature-dependent hydrophilicity, the hydrogel microcube array can be used to monitor the change of temperature at a desired position in a microfluidic chip (Fig. 5e). When the temperature of the flowing water increases from 10 °C to 50 °C, the hydrogel microcubes gradually shrank, resulting to the blue shift of their structural color, forming unique "color code" at each temperature (Fig. 5f–h). The reversible transformation can be repeated for many cycles (Fig. 5i). Such "color code" allows to quickly measure the temperature under microscope. The interference from the environment illumination or colored solutions, which is insurmountable for normal colloidal crystal sensors, can be avoided in our system by color calibration using the color information from different cubes. Utilizing the same manner, a temperature tag could be constructed, as shown in Supplementary Fig. 16, and a colloidal crystal hydrogel tag containing 14 connected cubes were fabricated. When the temperature rises, the color of the hydrogel tag will gradually vanish like a progress bar. Thus, we can directly read the qualitative temperature, which is much more convenient compared to hue analysis or reflection spectra measurement.

## Discussion

In summary, we proposed a sacrificial-scaffold-mediated TPL strategy that combined the bottom-up nanoparticle self-assembly process with

the top-down TPL process to fabricate colloidal crystal microstructures with diverse compositions, geometries, and structural colors. We demonstrate the feasibility to overcome the disturbance effect of the femtosecond laser with a degradable hydrogel network, therefore enabling the possibility to fabricate highly precise and free-designed colloidal crystal materials by TPL. The color and function of the obtained colloidal crystal microstructures can be adjusted by varying the processing parameters and type of precursors, respectively. Our strategy enables the possibility to endow nanomaterials with precise and free-designed 3D architectures while maintaining the orderly arrangement of their tiny units, thus is expected to lead to a number of innovative applications not only in the field of photonic crystals but also in other aspects, such as nanophotonics, nanocatalysis, and nanointelligence.

## Data availability
The data that support the findings of this study are provided in the manuscript, and its supplementary information, or are available from the corresponding author upon request. Source data are provided with this paper.

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

## Acknowledgements
This work was supported by the National Key R&D Program of China (No. 2017YFA0700500), the National Science Foundation of China (No. 22002015, 52033002, 52003051), the Fundamental Research Funding from Jiangsu Province (BK20211560), and the China Postdoctoral Science Foundation (No. 2022M710688, 2021M690617). X.D. thanks for the financial support from Jiangsu Key Laboratory of Oral Diseases (JSKLOD-KF-1903). Y.Z. thanks the financial support from the outstanding postdoctoral project of Jiangsu province (2022ZB126).

## Author contributions
X.D. and Z.G. conceived the research. X.D. and K.L. designed the experiments. K.L., H.D., S.L., Y.N. and Y.Z. performed the experiments and data analysis. J.Z. contributed to Raman characterizations. K.L. and Y.Z. performed the experiments during manuscript revision. X.D., K.L. and Y.Z. wrote the paper. All authors discussed the results and commented on the manuscript.

## Competing interests
The authors declare no competing interests.
