## [Peer Review File · Nature Communications]

3D Printing Colloidal Crystal Microstructures via Sacrificial-scaffold-mediated Two-photon LithographyReviewers' Comments:

Reviewer #1:

Remarks to the Author:

the manuscript presents a combination of high-resolution 3D printing and colloidal self-assembly as a means of defining mesostructured architectures with properties like photonics structural colors, etc. The paper is well written and the approach has interesting developments and elements of novelty, albeit colloidal particles of significantly smaller dimensions with more controlled geometry have been fabricated with the two-photon technique before. I think authors should better distinguish their approach from the past works on 2PP for colloidal systems (relevant works that come to my mind are Nature Materials 17, 71–79 (2018) and PNAS 112, 4546–4551 (2015) and NATURE MATERIALS 13, 258 (2014)), as well as cite the past works. They should also better emphasize what new possibilities their approach can bring for materials research and applications. In this respect, could one produce hierarchical colloidal particles and how small can these particles be? Could we then have these larger Microparticles self-assemble again? Can this approach be easily generalized to other colloidal systems, especially if metal nanoparticles were to be used as the small colloidal building blocks?

Reviewer #2:

Remarks to the Author:

The authors demonstrated a process that combines nanoscale 3d printing with self assembly. Here, silica nanoparticles assemble in a hydrogel matrix that is selectively cross linked to hold the particles in place. The work is rather similar to the recent publication here:

<https://pubs.acs.org/doi/10.1021/acs.nanolett.1c02483> with the following differences:

1. The particles seem to assemble while they are in the hydrogel solution
2. The particles are not removed after development
3. Various hues are achieved by varying degree of crosslinking.

Overall, I feel that the work is interesting and has elements of novelty. I would recommend that a detailed study of the mechanism that leads to color variation to be done. Some comments and questions below:

1. It's unclear to me how the particles can be made to assemble in a closed packed manner when they're in the hydrogel solution, while at the same time they're kept from moving about during laser exposure. Please explain this more clearly.
2. What is the resulting index of refraction of the matrix after curing? What is the index contrast between this and the particles.
3. Is it possible to further remove the particles to produce inverse opal structures?
4. Please provide more details on the cause of color change due to temperature variation. Would humidity have a role to play in this situation?
5. Overall, the manuscript could be improved for readability by improving the grammar and rearranging some of the results to be more concise.

Reviewer #1 (Remarks to the Author):

the manuscript presents a combination of high-resolution 3D printing and colloidal self-assembly as a means of defining mesostructured architectures with properties like photonics structural colors, etc. The paper is well written and the approach has interesting developments and elements of novelty, albeit colloidal particles of significantly smaller dimensions with more controlled geometry have been fabricated with the two-photon technique before. I think authors should better distinguish their approach from the past works on 2PP for colloidal systems (relevant works that come to my mind are Nature Materials 17, 71–79 (2018) and PNAS 112, 4546-4551 (2015) and NATURE MATERIALS 13, 258 (2014)), as well as cite the past works. They should also better emphasize what new possibilities their approach can bring for materials research and applications.

We thank the reviewer for the comment. We have cited and discussed the works referred by the reviewer. In addition, the potential possibilities and applications of our strategy is also discussed in the revised manuscript.

Following sentence was added in the revised manuscript:

Page 3, paragraph 1,

In fact, combination of the TPL and colloidal particle self-assembly have been demonstrated in some previous works, which enable the possibility to guide the alignment of nematic liquid crystals with colloidal particle geometries²⁴⁻²⁶, or generate inverse opal photonic crystal microstructures with colloidal particle templates²⁷.

Page 9, paragraph 1

Using our approach, photonic crystal hydrogels with arbitrary geometries can be generated on the target position of a substrate, this may enable the potential to construct complex and accurate sensors to monitor the change in motion and parameters of a microdomain in future.

Our strategy enables the possibility to endow nanomaterials with precise and free-designed 3D architectures while maintaining the orderly arrangement of their tiny units, thus is expected to lead to a number of innovative applications not only in the field of photonic crystals but also in other aspects such as nanophotonics, nanocatalysis and nanointelligence.

In this respect, could one produce hierarchical colloidal particles and how small can these particles be? Could we then have these larger Microparticles self-assemble again? We thank the reviewer for the comment. We prepared a series of hierarchical colloidal particles using our strategy (Fig. S10), the particles were designed to be a three-quarter hemisphere shape to avoid them fall off the substrate during development. The smallest hierarchical particle we can generate is with the diameter of $\sim 4 \mu\text{m}$. Due to the time-cost fabrication process of TPL, it is difficult to generate a large number of particles for self-assembly. However, we can directly fabricate an assembled array of hierarchical colloidal particles in hexagonal close-packed state (Fig. S10b). The array shows bright structure color (Fig. S10c), indicating the well-ordered arrangement of the nanoparticles inside.

Fig. S10 was added in the revised manuscript:

Fig. S10 a The SEM image of freeze-dried hierarchical colloidal particles with different feature sizes. The smallest hierarchical particle that can be generated is with the

diameter of $\sim 4 \mu\text{m}$. **b** The SEM image and **c** the optical image of a hierarchical colloidal particle array in hexagonal close-packed state.

Following sentences were added into the revised manuscript:

Page 9, paragraph 1

The generation of hierarchical colloidal particles and their assembled arrays were also achieved using our method (Fig. S10).

Can this approach be easily generalized to other colloidal systems, especially if metal nanoparticles were to be used as the small colloidal building blocks?

We thank the reviewer for the comment. We also look forward to expand our strategy to other colloidal systems, especially metal and metal oxide nanoparticles. Therefore, we tested this possibility using gold nanoparticles. The gold nanoparticles (140 nm diameter) are dispersed in BMOD-DMF solution to prepare the degradable hydrogel scaffold, which is then processed by TPL after filling with PEGDA precursor solution. However, problems arise in this process: when the concentration of the gold nanoparticles is low, microhydrogel could be obtained after degradation process, but the gold nanoparticles are not orderly distributed, probably because the interaction between the particles are too weak in this case (as shown in the figure below); when the concentration of the gold nanoparticles is high, the hydrogel scaffold cannot be degraded, probably due to the formation of a large amount of S-Au bond, which makes the gold nanoparticle as an undegradable crosslinker for the hydrogel scaffold. We believe that microhydrogels with ordered metal nanoparticles can be fabricated if we use another type of degradable hydrogel system instead of sulfur derivatives. We are currently working on this aspect and the results (if available) will be published in future.

a The SEM image of the fabricated microhydrogel cube containing 1.5 wt% gold nanoparticles. **b** The magnified SEM image showing the disorderly-distributed gold nanoparticles. **c** The gold element distribution map and **d** total element spectrum of the microhydrogel cube, obtained by energy dispersive spectrometry analysis.

Reviewer #2 (Remarks to the Author):

The authors demonstrated a process that combines nanoscale 3d printing with self assembly. Here, silica nanoparticles assemble in a hydrogel matrix that is selectively cross linked to hold the particles in place. The work is rather similar to the recent publication here: <https://pubs.acs.org/doi/10.1021/acs.nanolett.1c02483> with the following differences:

1. The particles seem to assemble while they are in the hydrogel solution
2. The particles are not removed after development
3. Various hues are achieved by varying degree of crosslinking.

We thank the reviewer for the comment. Indeed, the work referred by the reviewer is similar with our paper to some extent, we have cited and discussed this paper in the revised manuscript. In our opinion, the two papers belong to two different strategies commonly used when preparing photonic crystal materials.

The first strategy (corresponding to the ACS Nano paper referred by the reviewer) is to use an assembled colloidal particle film as template. Then, the precursor solution is poured into the templates and cured. Photonic crystal material (normally called inverse opal structure) can be obtained after removing the template. The advantage of this strategy is that the assemble of the colloidal particles would not be affected by the type of the secondary-added material, thus various materials can be utilized to generate photonic crystals. The disadvantage of the strategy is that creating a thick template is quite time-consuming, therefore the height of the fabricated inverse opal structures is normally limited. In addition, when soft hydrogels are fabricated by this strategy, the obtained structural colors are usually indistinct, as the inverse opal structure is easily deformable in this case.

The second strategy (corresponding to our paper) is to form a precursor solution containing colloidal particles and monomers. When the concentration of the colloidal particles in the solution is high enough, the particles can assemble to form well-ordered structures in the solution. Photonic crystal hydrogels can be directly obtained after curing the precursor solution. The advantage of this strategy is that thick photonic crystal hydrogel films can be quickly generated as there is no need to fabricate the colloidal template. The disadvantage of the strategy is that the assembly of the colloidal particles in the solution is very weak compare to the assembly in solid state, it can be easily disturbed by heat, vibration and existence of additional positive/negative-charged molecules in the solution. In our paper, we developed a sacrificial scaffold to “lock” the assembled colloidal particles in the solution, thus avoid the disturbance of their arrangement by the monomers and the TPL process.

Following sentence was added in the revised manuscript:

Page 3, paragraph 1,

In fact, combination of the TPL and colloidal particle self-assembly have been demonstrated in some previous works, which enable the possibility to guild the alignment of nematic liquid crystals with colloidal particle geometries²⁴⁻²⁶, or generate inverse opal photonic crystal microstructures with colloidal particle templates²⁷.

Overall, I feel that the work is interesting and has elements of novelty. I would recommend that a detailed study of the mechanism that leads to color variation to be done. Some comments and questions below:

1. It's unclear to me how the particles can be made to assemble in a closed packed manner when they're in the hydrogel solution, while at the same time they're kept from moving about during laser exposure. Please explain this more clearly.

We thank the reviewer for the comment. As we referred above, when the concentration of the colloidal particles in the solution is high enough, the particles can assemble to form well-ordered structures in the solution. We first made a precursor solution containing high ratio of SiO₂ nanoparticles to allow them assemble in the solution. Then we “lock” this ordered arrangement by turning the precursor solution into hydrogel state. The positions of the nanoparticles are immobilized by the highly crosslinked (but degradable) hydrogel network, thus they can hardly move during the filling of the secondary hydrogel precursor solution and the TPL process. We have explained the fabrication process more clearly in the revised manuscript.

Following sentences were added into the revised manuscript:

Supporting information, page 2, paragraph 2

During this process, secondary polymerization is induced by the high-energy laser in the degradable scaffold, while the arrangement of the nanoparticles is “locked” by the highly cross-linked degradable network. In this way, micro hydrogels with presetted geometries can be generated in the sacrificial scaffold by TPL without disturbing the ordered assembly of the inner nanoparticles.

2. What is the resulting index of refraction of the matrix after curing? What is the index contrast between this and the particles.

We thank the reviewer for the comment. The refractive index of the cured matrix is closely related to the solvent used in the precursor liquid and the proportion of the solvent. Taking the PEGDA hydrogel used in the article as an example, the refractive

index of the cured hydrogel matrix is 1.427, measured by Abbe refractometer. And the refractive index of silica particles is about 1.46. We have added this information in the revised manuscript.

Following sentence was added in the revised manuscript:

page 4, paragraph 3

This was confirmed by our test, as shown in Fig. S1a, a colloidal crystal hydrogel film with blue structural color can be easily obtained by curing an aqueous precursor containing polyethylene glycol diacrylate (PEGDA, the refractive index of the cured hydrogel matrix is 1.427) and silica nanoparticles ($\Phi=150$ nm, refractive index 1.46) under the UV irradiation.

3. Is it possible to further remove the particles to produce inverse opal structures?

We thank the reviewer for the comment. As SiO_2 can be dissolved in hydrofluoric acid (HF) solution, while polymers do not, it is possible to form inverse opal structures by our method. We have tested this potential in the revised manuscript. To do this, we chose zirconia ceramic instead of glass as the substrate since the tolerance of zirconia ceramic in HF is much better than that of glass, which slow down the detachment of the microstructure from the substrate during the etching of silica nanoparticles. We prepared the a microcube using PTTA as precursor solution, and immersed the cube in 5 wt% HF aqueous solution for 12 hours. Then, the samples were lyophilized in a freeze dryer. It can be seen from the Fig. S14a that the etched microstructures still adhere to zirconia substrates. Magnified SEM images clearly shows the inverse opal structure on the microcube.

Fig. S14 was added in the revised manuscript:

Fig. S14 **a** The SEM image of the inverse opal microstructure adhered to zirconia substrate. **b** The morphology of residual cross-linked network.

Following sentences were added in the revised manuscript:

page 10, paragraph 1

3D micro-object with more complex structure (Fig. 5c) and inverse opal microstructures (Fig. S14) were also fabricated, showing the ability of our method to create solid-state colloidal crystal materials with desired geometries.

Supporting information, page 3, paragraph 1

Fabrication of inverse opal microstructures: the degradable colloidal crystal hydrogel film was prepared on zirconia substrate. Next, the hydrogel was infused with the PTTA precursor in the dark. Then, TPL was performed, and microstructures were obtained after degradation. The microstructures were placed in 5 wt% hydrofluoric acid aqueous solution for 12 hours, which etches silica nanoparticles to obtain inverse opal microstructures.

4. Please provide more details on the cause of color change due to temperature variation. Would humidity have a role to play in this situation?

We thank the reviewer for the comment. We apologize for the misleading description of the experiment. The hydrogel is completely immersed in water in a microfluidic chip, thus the humidity would not affect the hydrogel color in this case. In the experiment, N-isopropylacrylamide is used as a temperature-responsive monomer to construct the temperature-sensitive photonic crystal hydrogel. As the temperature increases, the

hydrogel deswells, leading to the decrease of the nanoparticle spacing and therefore blue-shift of the structural color. We have added the photos of the experimental setup in the revised manuscript and corresponding description in the experimental part.

Fig. 5 is updated:

Fig. 5 Colloidal crystal materials with various compositions generated with the sacrificial scaffold. **a** Colloidal crystal hydrogel films fabricated with different precursors. The upper is the structure of the monomers and the bottom is the fabricated hydrogel film (left: without sacrificial scaffold; right: with sacrificial scaffold). With the sacrificial scaffold, hydrogel films with bright structure color were obtained in spite of the compatibility of the monomers with nanoparticles. Scale bar: 3 mm. **b** Solid-state colloidal crystal microcube fabricated using pure PTTA as precursor. **i** Model of the microcube ($L = W = 50 \mu\text{m}$, $H = 25 \mu\text{m}$). **ii** SEM image of the freeze-dried hydrogel microcube. **iii** Photo of the cube in DMF and **iv** in water. Scale bars: $25 \mu\text{m}$. **c** Solid-state colloidal crystal 3D microstructure fabricated with pure PTTA as precursor. **i** Model of the gyroid microstructure. **ii** SEM image of the obtained gyroid microstructure. **iii** Top view of the gyroid microstructure. **iv** 3D view of the object. Scale bar: $20 \mu\text{m}$. **d** Processing parameter of the temperature-sensor based on poly(NIPAAm) microhydrogel array. **e** Photo of the temperature-sensing array encapsulated in a microfluidic chip. Scale bar: 1 cm (top), 0.5 cm (bottom left) and $50 \mu\text{m}$ (bottom right),

respectively. **f** Photo of the hydrogel array under different temperature. Scale bar: 50 μm . **g** Reflective spectra of the hydrogel cube 1 under different temperature. **h** The maximum range of the reflection peak change of the hydrogel cubes under different temperature. **i** Change of the reflection peak of hydrogel cube 1 and cube 9 when temperature switch between 20 $^{\circ}\text{C}$ and 40 $^{\circ}\text{C}$ for 10 cycles.

Following sentences were added in the revised manuscript:

page 11, paragraph 1

Since poly(NIPAAm) exhibit temperature-dependent hydrophilicity, the hydrogel microcube array can be used to monitor the change of temperature at desired position in a microfluidic chip (Fig. 5e). When the temperature of the flowing water increases from 10 $^{\circ}\text{C}$ to 50 $^{\circ}\text{C}$, the hydrogel microcubes gradually shrank, resulting to the blue shift of their structural color, forming unique “color code” at each temperature (Fig. 5f-h).

Supporting information, page 4, paragraph 4

Temperature sensing experiment: the temperature-sensitive microsensor was fabricated on a glass substrate, which is then encapsulated by a PDMS cover to form a microfluidic chip. During the experiment, the microsensor was completely immersed in water, and the ambient temperature of the microsensor can be changed by pumping water with different temperatures into the microfluidic channel.

5. Overall, the manuscript could be improved for readability by improving the grammar and rearranging some of the results to be more concise.

We thank the reviewer for the comment. In the revised manuscript we have carefully examined the text and corrected all errors we found.

Reviewers' Comments:

Reviewer #2:

Remarks to the Author:

The authors have addressed all concerns raised satisfactorily

Reviewer #2 (Remarks to the Author):

The authors have addressed all concerns raised satisfactorily

We thank the reviewer for the comment.